# Evidence of torpor in the tusks of *Lystrosaurus* from the Early Triassic of Antarctica

Megan R. Whitney [1][✉] & Christian A. Sidor [2]

Antarctica has hosted a wide range of ecosystems over the past 500-million years. Early in the Mesozoic, the Antarctic portion of southern Pangaea had a more habitable climate, but its position within the polar circle imposed extreme photoperiod seasonality on its resident flora and fauna. It remains unclear to what degree physiological adaptations underpinned the ability of tetrapods to establish the terrestrial communities captured in the fossil record. Here we use regular and stressful growth marks preserved in the dentine of ever-growing tusks of the Early Triassic mammalian predecessor, *Lystrosaurus*, to test for adaptations specific to this polar inhabitant. We find evidence of prolonged stress indicative of torpor when compared to tusk samples from non-polar populations of *Lystrosaurus*. These preliminary findings are to our knowledge the oldest instance of torpor yet reported in the fossil record and demonstrate unexpected physiological flexibility in *Lystrosaurus* that may have contributed its survivorship through the Permo-Triassic mass extinction.

[1] Museum of Comparative Zoology, Harvard University, Cambridge, MA 02138, USA. [2] Department of Biology and Burke Museum, University of Washington, Seattle, WA 98195, USA. [✉]email: meganwhitney@fas.harvard.edu

Antarctica is today the coldest and driest continent with extreme variation in light availability throughout the year, restricting vertebrate life to coastal regions and rendering most of the continent uninhabitable[1]. These modern environmental conditions are anomalous, however, considering the deep history of life in Antarctica when flora and fauna occupied large regions of the continent[2]. Although more habitable with a warmer climate than today, Antarctica remained at a high latitudinal position for much of the Phanerozoic[3], subjecting its inhabitants to extreme photoperiod seasonality[4–6]. Extant vertebrates living in highly seasonally variable climates have evolved a variety of mechanisms to curb the effects of regular intervals of stress including daily torpor, hibernation, and brumation[7]. These adaptations are largely behavioral thus rendering them difficult to study directly in the fossil record. Importantly, however, these adaptations reflect underlying metabolic changes in response to resource limitations, and therefore should be recognizable in fossil hard tissues that preserve chronological records of physiology[8–13].

Geological data from the Early Triassic document prolonged and unfavorable environments following the Permo-Triassic mass extinction (PTME)[14]. PTME survivors and newly evolved species would have had to adapt to high global temperatures, oceanic anoxia, and low nutrient availability that created unstable and highly variable environmental conditions[15]. Polar regions are thought to have shielded their inhabitants from the extremes of these conditions both during the PTME and subsequently in recovery. The Fremouw Formation of Antarctica provides some of the earliest records of terrestrial vertebrates of Early to Middle Triassic age (~250–230 million years old)[16,17]. Furthermore, the flora and fauna of the Fremouw Formation is taxonomically similar to those found in non-polar regions of southern Pangaea, especially the Karoo Basin of South Africa[17–19], facilitating direct comparisons of polar and non-polar populations.

Here, we compare the frequency and patterns of growth marks in tusks of the Early Triassic non-mammalian synapsid *Lystrosaurus*, from polar Antarctica to those from the non-polar Karoo Basin of South Africa. Dentine, a major tissue of the vertebrate dentition, is deposited during times of regular incremental growth as well as times of arrested growth reflecting metabolic stress (Fig. 1). Compared to bone or enamel, dentine acts as a particularly sensitive recorder of daily-to-monthly physiological activity providing a robust chronology of both regular and stressed growth and has previously been used to assess responses to environmental change[8–13,20]. Despite this, the lack of sufficient experimental data on dentine deposition in extant tetrapods makes specifying the absolute amounts of time recorded by each growth mark (i.e. daily growth, yearly growth, etc.) difficult. Instead, we adopt an agnostic terminology reflective of growth patterns where fine-scale, regular marks denote baseline or routine growth and thicker more pronounced growth marks are referred to as stress marks.

The tusks of *Lystrosaurus* serve as particularly extensive markers of seasonality because they have been shown to be ever-growing[21]. Such tusks can therefore capture extended periods of time and control for the developmental stage of the tooth, reducing any bias toward rapid growth in newly erupted teeth. Furthermore, ever-growing dentitions such as incisors of modern rodents are known to preserve evidence of seasonal stress and hibernation[8–11] as well as in fossil rodents[12] and the tusks of mammoths[13]. Previous descriptions of a "hibernation zone" in these dentitions are recorded as episodes of shortened intervals between stress lines[8,9,12] and as such, we accordingly employ these characteristics in testing for torpor in our Early Triassic sample.

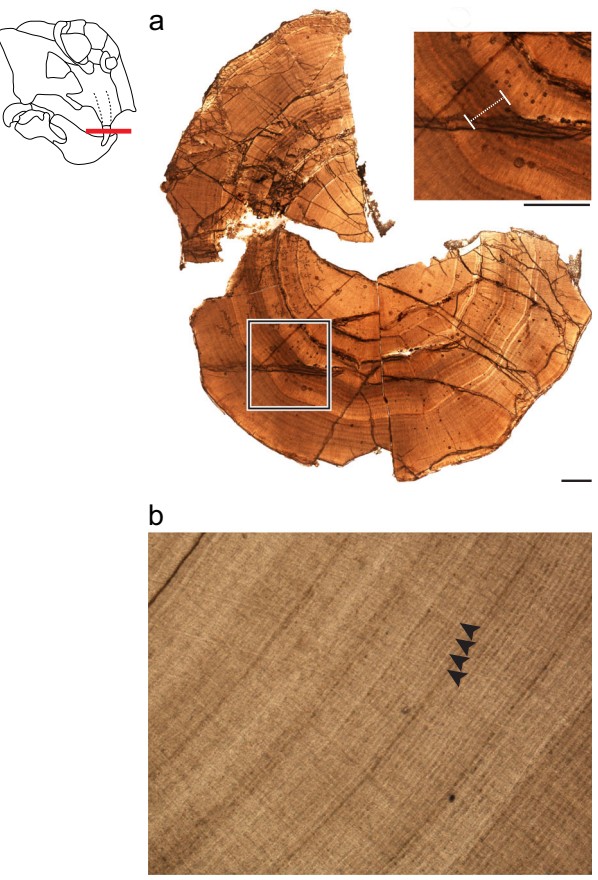

**Fig. 1 Measurements for stress and regular growth were recorded from the tusks of *Lystrosaurus*. a** A cross-section of Antarctic specimen UWBM 118025 with a "hibernation zone" highlighted at a higher magnification. Scale bars = 1000 μm. **b** Well-preserved regular incremental growth marks from the South African specimen UWBM 118028, lacking "hibernation zones". Arrows denote individual lines with an average spacing of 16–20 μm. Scale bar = 100 μm.

We find evidence of prolonged and repeated metabolic stress events in the tusks of Antarctic *Lystrosaurus* that differ from the relatively steady growth observed in South African *Lystrosaurus* tusks. The patterns of stress found in polar *Lystrosaurus* are similar to previously described "hibernation zones" suggesting that Antarctic *Lystrosaurus* experienced seasonal torpor, likely similar to hibernation. This preliminary finding supports the growing body of evidence that *Lystrosaurus* was endothermic and highlights the role of polar regions in the recovery of terrestrial vertebrates from the PTME.

## Results

**Regular growth marks**. We found that the spaces between regular, incremental growth marks in the tusks of Antarctic *Lystrosaurus* were not significantly different from those of South African *Lystrosaurus* (Fig. 2) indicating that although geographically separated by over 900 km in the Triassic, the baseline physiology and growth was generally similar in both populations and across individuals in our sample. These baseline indicators of physiology allow us to control for alternative sources of variation in growth patterns between the two populations (i.e. differences in growth rate due to ontogeny or species/individual variance). In modern ever-growing dentitions, these regular growth marks have been demonstrated to vary with factors such as age[22], species of varying metabolisms[23], and dentine deposition is sensitive to

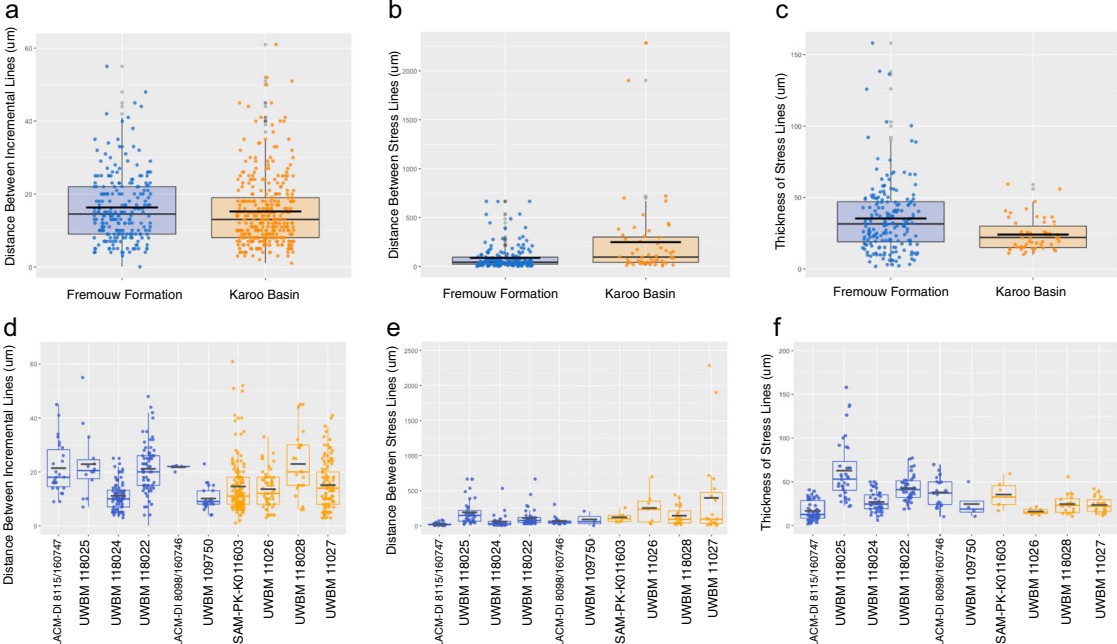

**Fig. 2 Stress and regular incremental data for Antarctic (blue) and South African (orange) *Lystrosaurus* tusks. a** The mean distance between regular incremental growth marks was insignificantly different between the two populations ($p = 0.06543$; $n = 550$) suggesting similar baseline metabolic activity in Antarctic and South African *Lystrosaurus* populations. **b** The distance between growth marks was significantly greater in South African specimens ($p = 7.43e-05$; $n = 234$) indicating longer durations of uninterrupted growth. **c** The thickness of stress lines in Antarctic specimens was significantly greater than South African specimens ($p = 1.45e-03$; $n = 246$) indicating longer periods of inactive growth. **d–f** Measurements averaged in **a–c** separated by data collected for each specimen. Considerable variation was observed, but even with outliers removed, the average significant differences remained consistent.

nutrient inputs[24]. Completely controlling for alternative sources of variation was particularly difficult given the small and fragmentary nature of the specimens that were available for destructive sampling, however, we consider these regular growth marks to be at least a general control for those sources of variation in growth that are unrelated to seasonality.

**Stress marks**. While our sample contained individuals of similar baseline metabolic activity, indicators of metabolic stress were different between Antarctic and South African specimens. Quantitatively, both the duration of stress and the short intervals between stressful events suggest that Antarctic *Lystrosaurus* experienced relatively frequent and pronounced metabolic strain (Fig. 2). The amount of growth between stressful events was, on average, significantly greater in South African *Lystrosaurus* suggesting longer durations of regular growth without stressful interruptions for non-polar populations that inhabited the Karoo Basin. Furthermore, stressful events appear to have lasted for longer durations in Antarctic fossils as suggested by the significantly thicker lines observed in tusks from the Fremouw Formation. Antarctic tusks preserve stressed growth marks akin to previous descriptions of a "hibernation zone" where a series of thick stress marks are found close to one another while South African specimens typically display a single stress mark followed by regular growth that constitutes the majority of the growth record of this population (Fig. 3 and Supplementary Fig. 1).

**Intrapopulation variation**. Substantial variation within localities does exist. However, even when outliers are removed from our quantitative analyses, statistically significant differences in stress marker measures persist. In fact, this variation is an important consideration given that not every stress line necessarily represents torpor, even in the Antarctic populations (Fig. 4). Short periods of metabolic reduction may be caused by a variety of

factors, but the unique patterns here observed in Antarctic tusks, where occurrences of closely spaced stress lines are present, are consistent with torpor at high latitudes.

## Discussion
From this exploratory study, we find evidence of severe and prolonged periods of stress in Antarctic *Lystrosaurus* tusks that support the conclusion that polar populations adapted to their high-latitude environment by means of seasonal reduction in metabolic activity, otherwise referred to as torpor. The "hibernation zones" denoted here by temporarily reduced dentine deposition between stress lines are quantitatively (Fig. 2) and qualitatively (Fig. 3 and Supplementary Fig. 1) akin to heterothermic activity observed in modern endotherms. Heterothermic ectotherms reduce their metabolic activity from at least half of to nearly complete quiescent metabolic activity[25]. Heterothermic endotherms, on the other hand, can enter a state of torpor generally reducing metabolic activity by at most a third although most reduce by no more than 10% of normal activity[7]. Generally, ectothermic heterotherms are not able to reactivate metabolic activity during unfavorable environmental conditions and enter times of brumation whereas endothermic heterotherms, even hibernators, frequently will come out of metabolic dormancy either daily, weekly, or monthly[26]. The zones of stress observed here in Antarctic *Lystrosaurus* are marked by iterative reactivation of metabolic activity similar to those seen in torpor patterns of modern endotherms. This contributes additional support for a growing body of evidence that dicynodonts like *Lystrosaurus* were likely endothermic[27–30].

These data also shed light on Antarctica's role as a refugium during the PTME. Antarctic rocks have yielded Early Triassic tetrapod taxa that are missing from other contemporaneous, but otherwise much better sampled localities such as the Karoo Basin of South Africa. These discrepancies in otherwise very similar

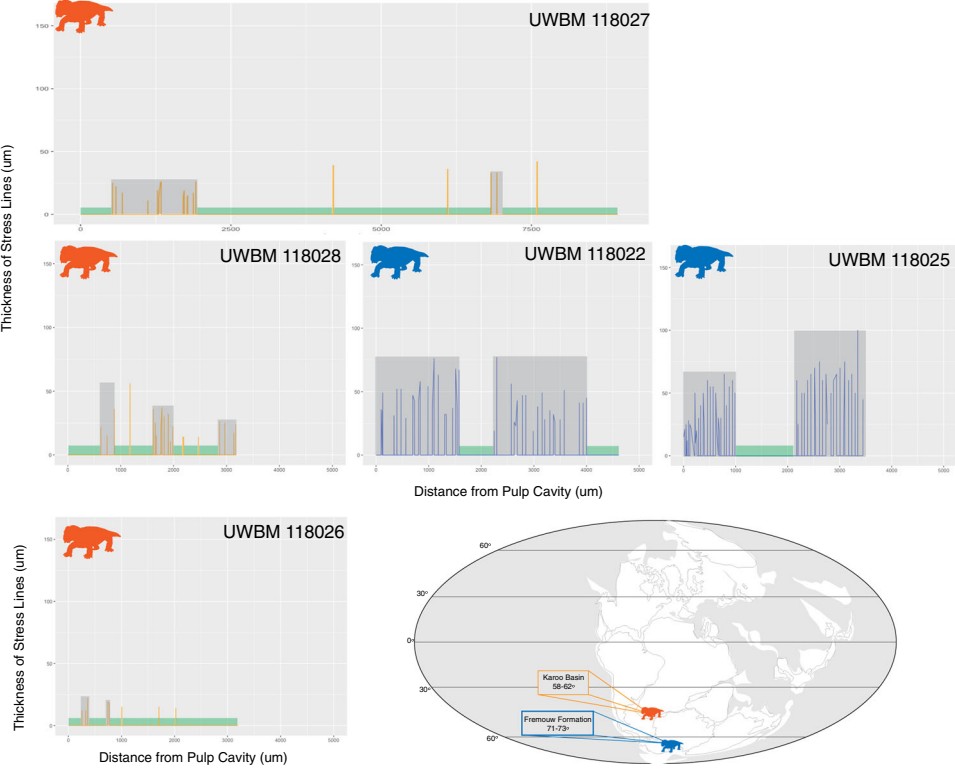

**Fig. 3 Summary of stress mark data from a selection of South African (orange) and Antarctic (blue) specimens of *Lystrosaurus*.** These graphs represent transects collecting the thickness (μm) of stress lines moving from the pulp cavity (to left) to the outer edge of the tusks (to right). Green boxes highlight durations without visible stress lines and gray boxes highlight portions of the tusk with closely spaced stress lines suggesting a stressful interval. The height of the boxes represents the thickest stress line during such period. See Supplementary Fig. 1 for transects from entire sample.

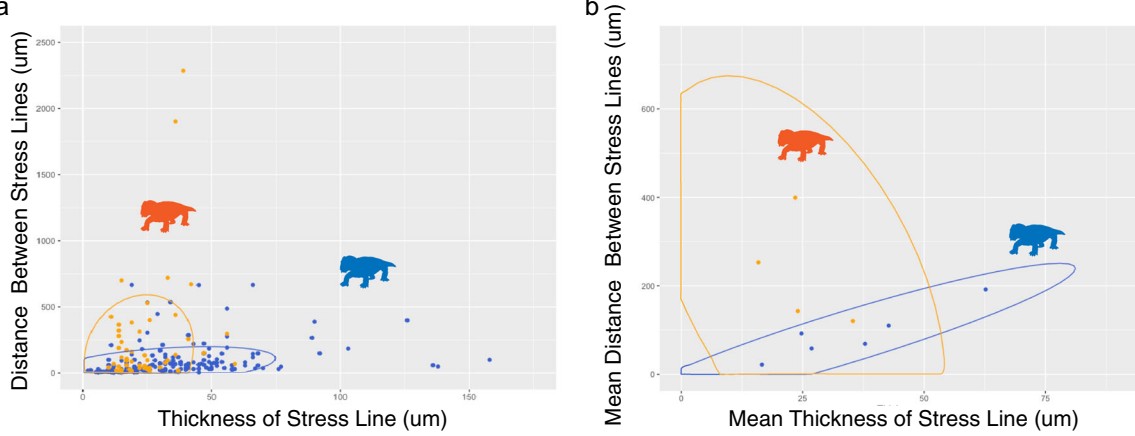

**Fig. 4 Comparative measurements of stress lines in Antarctic (blue) and South African (orange) tusks.** There is substantial variability within each population, however, South African stress lines tend to be thinner and farther apart than Antarctic specimens which are more likely to occur closer together and more frequently. This is apparent as individual data points (**a**) and when specimens are assigned a mean value for these measurements (**b**).

faunal assemblages[17–19], have supported the hypothesis that the Antarctic portion of Pangaea was a high-latitude refugium from the global climatic events marked by the PTME[31,32]. Furthermore, geologic data suggest that polar regions were, in fact, the first to begin a prolonged recovery in the Early Triassic[15]. With its relatively temperate climate, Antarctica may have acted as a haven for terrestrial vertebrates through an extinction boundary and subsequent recovery.

Although more insulated from the effects of global climate change, tetrapods living in Antarctica during the Early Triassic would have had to adapt to extreme seasonality with long periods of limited light availability. A Permian tetrapod assemblage has yet to be recorded from Antarctica[33], however, *Lystrosaurus* tusks from both the Permian and Triassic of the Karoo Basin do not record patterns of hibernation-like reductions in their metabolic activity (Fig. 3). Thus, these data suggest that upon expanding its geographic range to the Antarctic portion of southern Pangaea *Lystrosaurus* adapted with extended periods of reduced metabolic activity, although continued testing of this initial observation is required.

*Lystrosaurus* survivorship through the PTME and its subsequent abundance in the fossil record of the earliest Triassic[34] was likely predicated on a flexible physiology that could modulate typically elevated metabolic activity according to the limiting resources of a fluctuating environment[34,35]. This agrees with Valentine's[36] suggestion that more stable environments tend to select for narrow niche partitioning among species, whereas those with unstable resources, such the Early Triassic[15], tend to select for species with greater flexibility and generalization. Indeed, the near-global distribution of *Lystrosaurus*, with records known from China, Russia, India, Africa, and Antarctica, implies a remarkable ecological breadth for this lineage[37]. Furthermore, Triassic rocks from both South Africa and Antarctica preserve fossil evidence of tetrapod burrowing[38–40], including for *Lystrosaurus*[38]. We suggest that a combination of a flexible physiology and burrowing served as exaptations to the acquisition of torpor for Antarctic populations of *Lystrosaurus*.

We show preliminary evidence that *Lystrosaurus* used torpor to respond to the seasonal stress incurred specifically in the polar regions of southern Pangaea during the Early Triassic. It was this ability to sustain activity in a variety of stressful environmental conditions that may have served as a critical adaptation in surviving and recovering from the largest mass extinction the earth has experienced to date. The Fremouw Formation preserves a diverse assemblage of tetrapod taxa[17–19] with presumably an array of metabolic adaptations to the extremes of seasonal light availability. Continued testing of seasonal responses in *Lystrosaurus*, other non-mammalian synapsids, early reptiles, and the abundant temnospondyl fossils recovered from these localities can reveal how this polar ecosystem evolved despite unstable environmental conditions and eventually facilitated post-extinction recovery.

## Methods

**Overview**. Following standard thin-sectioning protocol established by Lamm[41], transverse and longitudinal sections of six Antarctic and four South African *Lystrosaurus* tusks were made and analyzed (Supplementary Table 1). Paleolatitude was estimated from tectonic plate reconstructions using paleomagnetic reference frame developed by Torsvik et al.[42] and made available through the online calculator developed by van Hinsbergen et al.[43].

**Dentine growth mark terminology**. Generally, there are three kinds of growth marks recognized in dentine: (1) daily increments that correspond to a circadian rhythm called lines of von Ebner, (2) 6–10 day increments called Andresen lines, and (3) periods of stress where dentine deposition essentially ceases creating particularly thick growth marks[44,45]. While these three marks are recognized in primates, especially humans, there is little investigation into the consistency of this hierarchy outside of mammals. This is in large part due to the range of reported distances between these growth marks. Dean[44] reviewed the discrepancies in both reported short periods between lines of von Ebner (varying from 1.7 to 20 μm) and long periods between Andresen lines (varying from 4 to 20 μm) for which values overlap. Erickson[46] was one of the few to experimentally test for daily lines of von Ebner in a non-mammalian vertebrate (*Alligator mississipiensis*). Erickson, however, only reported that lines of von Ebner were spaced <20 μm apart and did not discuss the variation of space between lines or whether or not there was a distinction between short- and long-term periods. Furthermore, that study was conducted on juvenile alligators that were rapidly growing and thus reported distances between growth marks may not be truly comparable to the adult primates previously studied or representative of adult dentine growth more generally. In this study, we follow Dean's recognition of short-term and long-term period growth marks generally as regular incremental lines, assigning no specific amount of time, but rather that they represent routine or baseline dentine deposition (Fig. 1).

**Statistics and reproducibility**. An additional aim of this study was to develop a more quantitative and objective method for measuring and counting growth marks, especially in fossil material where visuals can be impaired by taphonomic alteration. Using NIS Nikon Elements software ROI Measurement Tool, we ran transects across live images of tusks that collected measurements on light intensity. The output of these transects records decreases and increases in light intensity that can be confirmed in the live image feed as growth marks and spaces between growth marks are recorded. Relative dips in light intensity are associated with regular and stress lines whereas relative increases in light intensity represent times

of growth and tissue deposition. The space between regular increments ($n = 550$) as well as distance between ($n = 234$) and thickness of stress intervals ($n = 246$) were captured in these light transects. Mann–Whitney $U$ tests were used to determine statistical significance in these nonparametric data sets (Fig. 2). Ellipses were constructed assuming a multivariate t-distribution at 95% confidence intervals (Fig. 4).

**Reporting summary**. Further information on research design is available in the Nature Research Reporting Summary linked to this article.

## Data availability

Thin sections of UWBM specimens (118022, 118024, 118025, 118026, 118027, 118028, and 109750) are housed at the University of Washington Burke Museum, thin sections and additional material of LACM-DI 8115/180747 and LACM-DI 8098/160746 are housed at the Natural History Museum of Los Angeles County and thin sections and additional fossil material of SAM-PK-K011603 is housed at the Iziko: South African Museum. Histological images that support this study have been deposited in MorphoBank with the project number 3650 at http://morphobank.org/permalink/?P3650.

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

## Acknowledgements

Funding for this project came from NSF PLR-1341304 (awarded to C.A.S.) and NSF DEB-1701383 (awarded to M.R.W.). Y. Tse, A. Huttenlocker, and K. Button assisted in specimen preparation. We thank Z. Skosan (Iziko: South African Museum) and M. Walsh (Natural History Museum of Los Angeles County) for access to specimens. We thank the 2017–2018 Shackleton Glacier field team (P. Braddock, P. Makovicky, J. McIntosh, A. Shinya, N. Smith, R. Smith, and C. Wooley) and W. Hammer for Antarctic field collections. Finally, logistical support for this project in Antarctica was provided by the U.S. National Science Foundation through the U.S. Antarctic Program and we thank the Shackleton deep field camp staff and pilots for supporting this research.

## Author contributions

Study design was developed by M.R.W. and C.A.S.; M.R.W. collected and analyzed histological data and both M.R.W. and C.A.S. contributed to the writing of this manuscript.

## Competing interests

The authors declare no competing interests.
