## [Peer Review File · Communications Biology]

Reviewers' comments:

Reviewer #1 (Remarks to the Author):

This study compares the dentine of Permo-Triassic *Lystrosaurus* tusks from Antarctica and South Africa to deduce aspects of the physiology of this genus. This non-mammalian synapsid was one of the few terrestrial taxa to survive the end-Permian Mass Extinction and the only terrestrial vertebrate to increase in abundance directly after the extinction event. It can thus act as an excellent case study for asking questions relating to organismal response to massive environmental perturbation such as that which occurred during the end-Permian Mass Extinction. Given the importance of *Lystrosaurus* in such studies, the authors made the right choice in choosing this taxon to test their hypothesis. The authors compared the dentine of Triassic *Lystrosaurus* tusks between a relatively polar environment (Antarctica) and a more temperate environment (South Africa) and quantified the differences. They propose that thicker, more prominent stress lines (exhibited when an animal undergoes some form of stress during growth resulting in decreased dentinal deposition) are present in the Antarctic specimens compared to those from the South African Karoo Basin. The authors suggest that these stress lines represent evidence of torpor in the individuals that were living in Antarctica during the Early Triassic and further propose that Antarctica acted as a refugium for this population of *Lystrosaurus*. These results are potentially very exciting. If shown to be accurate, this study represents the oldest example of torpor in a vertebrate and shows how a vertebrate such as *Lystrosaurus* could respond to changing climate conditions after a mass extinction (by initiating torpor probably through burrowing). The implications of these results are significant and globally relevant, and highly suitable for this journal. However, I have one major concern, which results in a number of questions being raised regarding the interpretation of the results.

1. Six Antarctic and only four South African specimens were used in this study (please also note that only five Antarctic specimens are listed in the table – this discrepancy needs to be fixed). It is very difficult to move beyond an observation to statistical significance with such a small sample size. Although each population has more than three tusks, there are a number of possible causes for the observed variation that need to be ruled out before linking the observations to torpor. These include (but not limited to) ontogenetic stage, inter-individual and inter-specific variability. For example, what are the implications of comparing teeth from animals at different ontogenetic stages? Although dentine is ever growing, as the authors know, it becomes worn away during the animal's life time, so we do not really know how old the animal was at the time of death (i.e. what ontogenetic stage was recorded in the teeth). Furthermore, it has recently been shown that none of the Triassic *Lystrosaurus* specimens from the South African Karoo Basin were fully grown, most of them being juveniles and young subadults (Botha-Brink et al. 2016, Scientific Reports). However, those from Antarctica have not been tested. If ontogenetic stage has little or no effect on the deposition of dentine, then the authors should use a reference to rule this out (I'm sure one could find a reference from the human literature) so that comparing juveniles with adults is not a problem. Additionally, I can see from the cross-section in the figure that the decrease in dentinal deposition is temporary and the distance between lines increases again after the "hibernation zone", but this is not stated in the text – one sentence will clarify that the decrease in dentine deposition is not due to age.

2. The authors note that considerable variation was observed, but there was still a significant difference between the two groups. The box plots show quite a noticeable overlap and with a sample size of 4 and 5/6 it is rather premature to consider these groups significantly different from one another. How do you rule out individual variation here?

3. Furthermore, the authors are using different species of *Lystrosaurus*. South African specimen UWBM 118026 is from the Permian for example (noted as being found below the Permo-Triassic

boundary in the table) and thus represents either *L. maccaigi* or *L. curvatus*. In contrast the other three Triassic South African specimens represent *L. maccaigi*, *L. curvatus*, *L. murrayi* or *L. declivis* (with specimens being from just above the boundary to the Katberg Formation, it is highly unlikely that even the Triassic specimens represent one species). As the authors don't mention the species I assume that the specimens were not preserved well enough to identify them to species level. However, it raises the question, what is the species effect? It may be negligible given that they are so closely related, but this needs to be stated and the reason given. The stress mark data for UWBM 118026 looks a bit different to the other two South African specimens given in Figure 2 for example.

4. In the extended data figure 1e UWBM 11027 is labelled blue (Antarctica), but it appears orange (SA specimen) in figure 1d, f and h, and is listed as South African in the table. This needs clarification.

5. UWBM 118028 is listed in the figures with the full accession number, but 118026 and 118027 are not where the 8 is missing. If 118028 can fit on the figure so can the others. This should be rectified.

Thus, I find these results very interesting, but the small sample size and supporting text – as it stands now - does not represent compelling evidence for torpor in Antarctic *Lystrosaurus*. If the authors can justify (even if it's just including more references to other studies) why the observed differences don't represent other possibilities – with such a small sample size – then the paper will be suitable for publication.

Reviewer #2 (Remarks to the Author):

(See attached file)

I reviewed carefully the manuscript of Whitney and Sidor entitled "Evidence of torpor in the tusks of *Lystrosaurus* from the Early Triassic of Antarctica". In this study, the authors compared the tusk histology of several specimens of *Lystrosaurus* from Antarctica and South Africa and found evidence of torpor in polar populations denoted by differences in the growth mark record.

This manuscript is really well written and illustrated and the interpretations are well supported by the provided data and citations. I only have minor comments that are reported on the attached edited manuscript.

I therefore recommend publication of this study in *Communications Biology*.

Reviewer #3 (Remarks to the Author):

(See attached file)

Whitney and Sidor have analysed "regular" and "stressful" growth marks in the dentine of tusks of Early Triassic dicynodont specimens of a mammal-like reptile, *Lystrosaurus* (Therapsida), to test for adaptations in a polar (Antarctic) environment. From this preliminary exploratory study using small samples, they claim to have found evidence of prolonged stress indicative of torpor in the polar samples, in contrast to results obtained from non-polar (South African) specimens of *Lystrosaurus*. They claim that this is the oldest instance of torpor reported in the fossil record.

This particular claim is novel, but it is based on small samples (n= 6 from Antarctica and n= 4 for South Africa). It is important to state that this is a preliminary exploratory study. Having said that, I consider that the evidence the authors present deserves to be published.

Regarding statistical analyses: In "Extended Data Fig 1 b" there are two distinct outliers, which would greatly affect a mean value.

The manuscript is most certainly in need of extensive rewording and amendments. To assist the authors it is absolutely necessary for me to send as an attachment a file named "WHITNEY LYSTROSAURUS ms" to guide the authors. There are just too many amendments to list here one by one. In my opinion I consider that once the proposed amendments are attended to, the manuscript could be deemed acceptable in principle. It is not publishable as it stands.

I recommend the addition of one more Figure (Fig. 3, or at least the incorporation of an additional image in Fig. 2), to plot "number of stress lines per tooth" against "Mean thickness of stress lines". If this kind of thing is done in the manner that I have shown in the document named "WHITNEY LYSTROSAURUS ms", the authors will be in a stronger position to support the claim that they have identified torpor. A graph of the kind that I recommend for Fig. 3 will serve to summarise data for all ten tooth samples simultaneously in one image, based on two variables.

The authors evidently question Erickson's claim of daily growth increments in extant alligators, and they make no reference to Thackeray's (1991) pioneering work on growth increments in a Permian South African dicynodont (*Diictodon*) closely related to *Lystrosaurus* which is a dicynodont that is found in both Permian and Triassic deposits in South Africa. Reference to Thackeray's work must be made, even if only to acknowledge that growth increments (essentially identical to the "regular" growth increments in *Lystrosaurus* studied by Whitney & Sidor) have previously been studied in a South African context. The authors can then go on to say that they adopt an "agnostic" approach, but it is essential that they precede this statement by referring to both Erickson and Thackeray, irrespective of their claims to have identified possible daily growth increments, in modern and fossil samples respectively.

I consider that this article, if published, will influence thinking in the field of palaeontology. Certainly it will be provocative, but it will be very good to stimulate debate.

The article will stimulate further research, using larger samples.

I congratulate the two authors.

Reviewer 1

We found Reviewer 1's comments to be extremely helpful and believe that addressing these comments has made our manuscript much improved in providing both clarity and transparency in the limited nature of our sample size. We have addressed the reviewers' comments following the numbers outlined in the review.

1. We accidentally left out a specimen in our table (UWBM 109730) and have added it. Thank you for catching this!

Specimen Number	Inferred Paleolatitude	Locality Description
UWBM 118022	72° S	Coalsack Bluff in the Beardmore Glacier region of the Transantarctic Mountains within 30 meters of the base of the Fremouw Formation
UWBM 118024	72° S	
UWBM 118025	72° S	
UWBM 109750	72° S	
LACM-DI 8115/160747	73° S	Collinson Ridge in the Shackleton Glacier region of the Transantarctic Mountains within 80 meters of the base of the Fremouw Formation
LACM-DI 8098/160746	71° S	McIntosh Ridge in the Shackleton Glacier region of the Transantarctic Mountains
UWBM 118026	61° S	approximately 3 meters below the Permo-Triassic boundary in the Bethulie District, Eastern Cape Province of South Africa
UWBM 118027	58° S	7 meters above the Permo-Triassic Boundary in the Namakwa District, Northern Cape Province of South Africa
UWBM 118028	62° S	5 meters above a major sandstone package of the basal Katberg Formation in the Nieu Bethesda District, Eastern Cape Province of South Africa
SAM-PK-K011603	62° S	approximately 22 meters above the boundary from the Nieu Bethesda District, Eastern Cape Province of South Africa

We agree there are many sources of variation that could influence overall growth rates that may have contributed to the differences we see in metabolic stress. Unfortunately, the destructive nature of our sampling protocol has limited us to specimens that were relatively fragmentary. For this reason, identifying the ontogenetic stage and species were not possible. To control for this, we analyzed regular growth marks in *Lystrosaurus* dentine as a measure for major differences in their metabolisms. This is by no means a perfect measure (ideally, we would have multiple sectionable long bones from both populations and/or tusks from recognizable skulls), but it at least demonstrates that these two populations generally had similar baseline physiologies. We have added text to emphasize that these measures of regular growth act as a control for variance between the two populations and cited additional studies experimentally testing the sensitivity of these regular increments to baseline physiology. (see lines 67-75).

We agree the temporary nature of these "hibernation zones" was not clearly stated in our initial submission and have added text to clarify. (see lines 59, 80).

2. We cannot completely rule out individual variation. However, we did not find significant differences in the regular incremental data suggesting that what is different about these two groups is how they are exhibiting stress. Hopefully the added text clarifies that point. By displaying all of our data we believe it demonstrates that there is variation within these two populations (Extended Data Fig.1), but that there is still a statistically significant difference in the average measures of stress recorded. These quantitative assessments are complementary to our quantitative observations that there were episodes of closely spaced stress lines in the Antarctic sample.

3. The available specimens did not preserve enough anatomy to assign to species-level identifications. This possible alternative source for variation was accounted for as well as we could with our sample, which was to test variation in regular incremental lines. Also, as noted above, we are unaware of any previous demonstration that the species of *Lystrosaurus* varied appreciably in their physiology.

4. UWBM 118027 is a South African specimen. Thank you for catching this mistake! The color has been corrected on extended data figure 1E.

5. Again this was an oversight and we have added 8s to both specimen numbers.

Reviewer 2

We appreciate the line edits and suggested citations. Below we note a few points of disagreement, but generally, these helpful edits were added to our revision.

Line 26: we cited our survey of hard tissue seasonal histology here in addition to later in the text.

Line 26: we agree this sentence could be more specific and have edited to reflect this

Line 46: we added citations

Lines 64-65: we made edits to address alternative sources of variation in growth patterns and to clarify that regular growth marks acted as a control for differences in “normal” metabolic activity.

Line 67: our sample size is described in the methods section; we believe it is more appropriate to describe sampling in the methodology.

Lines 74-75: This is a statement we are making as part of our conclusion. There is no study that we are aware of that directly measures growth patterns in ecto vs. endothermic dentine (although there should be!). However, it is similar to hibernation zones that have been previously described in endotherms and follows a pattern (that we describe in this paragraph) of endothermic torpor. To clarify this point, we have edited the opening line of the paragraph to reflect that we are interpreting these data rather than drawing on a previous study.

Line 84: we have added these citations

Line 109: we tried to summarize these findings as briefly as possible in the first sentence of this paragraph to stay within a reasonable word limit. We added the suggested citations to address the importance of this previous work.

Line 109: we edited the text accordingly

Line 148: we edited the text accordingly

Line 168: we edited the text accordingly

Reviewer 3

The reviewer made many line edits that provided important points for us to improve our manuscript. Many of these suggestions were incorporated into our revision and many were stylistic differences where we often favored the phrasing in our original draft.

Comments on title: By leaving in “Evidence of”, we believe our title better reflects the preliminary nature of our findings. We did not find evidence of torpor in South African tusks; that sample was used as a comparison for the Antarctic specimens.

In response to general comments:

-Outliers were removed and significant differences were still observed between the means. This was addressed (line 68) in the original submission.

-Figure 3: We added an additional Supplemental figure that summarizes the data. We made our own adjustments since we believe that plotting thickness of and distance between stress lines summarized the data more effectively. We have displayed both the individual data points and the mean values for each specimen since we think it is important to display variation.

Supplementary Figure 2. Comparative measurements of stress lines in Antarctic (blue)

and South African (orange) tusks. There is substantial variability within each population, however, South African stress lines tend to be thinner and farther apart than Antarctic specimens who are more likely to occur closer together and more frequently. This is apparent as individual data points (left) and when specimens are assigned a mean value for these measurements (right).

-We have now cited Thackery's important contribution and relevance to this study.

In response to line edits:

Line 7: we edited the text accordingly

Lines 8-9: we edited some of the text accordingly, but on other points we disagree. We did not include the term "mammal-like reptile" as this is a pre-cladistic misnomer for non-mammalian therapsids.

Line 14: we edited the text accordingly

Lines 18-19: we edited the text accordingly

Lines 29-30: we edited some of the text accordingly, however the sentence has also been reworked from Reviewer 2 comments.

Line 31: we edited the text to reflect the preliminary nature of this study.

Line 39: global temperatures were particularly high at this time, not just relatively high

Line 42: we edited the text accordingly

Lines 48-49: these are stylistic differences and we prefer our version. We are limiting the taxonomic names used here (e.g. dicynodont) to make the text widely accessible.

Line 51: we believe it is important to describe the level of detail that these increments can capture

Lines 52-57: we believe explicit discussions of these studies are better suited for the methods section but are included in the citations of this statement.

Lines 59-60: it is unclear why the reviewer recommends italicized terms here. We did not make these edits.

Line 61: because these tusks are ever-growing they do serve as an extensive marker.

Lines 63-64: we believe it is important to include the developmental control using ever-growing dentitions can provide and thus have left the text as is.

Lines 68-69: these edits are style differences, but we did add Early before Triassic for clarity.

Line 72: We find this sentence breaks up the flow of our writing. Change not made.

Lines 74-75: we edited the text accordingly

Line 76: Figure 3 is included as an extended figure? We have addressed this in extended figure 2.

Line 79: significant differences in averages are important as they refer to the statistics rather than just general differences

Line 80: we edited the text accordingly

Line 81: this is a stylistic difference and changes were not made

Lines 84-85: we believe, especially given our small sample size, it is important to address these outliers.

Line 88: these sentences were rearranged and rewritten for clarity.

Lines 116-117: this is a stylistic difference and changes were not made

Line 127: we believe that providing evidence emphasizes the preliminary nature of this study more than contend; *Lystrosaurus* was italicized

Line 128: this is a stylistic difference and changes were not made

Line 129: we edited the text accordingly

Lines 140-141 (Fig. 1 caption): we edited the text accordingly

Line 145 (Fig. 2 title): we edited the text accordingly

Lines 224-227 (Extended data fig. 1 caption): we edited the text accordingly

REVIEWERS' COMMENTS:

Reviewer #1 (Remarks to the Author):

I have read the rebuttal letter and revised manuscript and the authors have justified their results satisfactorily.

Reviewer #3 (Remarks to the Author):

I commend the authors for making major improvements to the first version of their article on growth increments in *Lystrosaurus* tusks from Antarctica and South Africa. There are still some outstanding matters, which can (I think) be easily addressed.

In the abstract they state "We find evidence of prolonged stress indicative of torpor when compared to tusk samples from non-polar populations of *Lystrosaurus*. These preliminary findings are the oldest instance of torpor reported in the fossil record and demonstrate unexpected physiological flexibility in *Lystrosaurus*".

To their credit, the authors are claiming "the oldest instance of torpor reported in the fossil record". However, at issue is the claim that this is confined to Antarctic samples, to the exclusion of South African (Karoo) "non-polar populations of *Lystrosaurus*".

In their rebuttal to Referee #3, they state "We did not find evidence of torpor in South African tusks". But this is questionable in relation to the new graph in "Supplementary Figure 2" (in response to my earlier recommendation of a bivariate plot to explore patterns for all 10 specimens together).

I am questioning the following with regard to Supplementary Fig 2 (RIGHT): Of the four South African specimens, two may not necessarily reflect torpor, but the other two fall close to or even within the range of values of polar samples associated with torpor. Is this understood correctly? If the authors agree with this interpretation, the title of their important article might potentially be amended as follows:

EVIDENCE OF TORPOR IN TUSKS OF LYSTROSAURUS FROM THE EARLY TRIASSIC IN GONDWANA (ANTARCTICA AND AFRICA).

We appreciate the continued efforts of both reviewers to improve our manuscript. Reviewer 2 suggested accepting the revised manuscript. Reviewer 3 made a number of additional suggestions, which we outline below in the order in which these comments were made.

Reviewer 3's comments on our revised manuscript largely focus on the notion that torpor could have also occurred in *Lystrosaurus* specimens collected in South Africa. The basis for Reviewer 3's interpretation that South African specimens also had torpor largely seems to come from the mean values presented in Supplementary Figure 2 (now Figure 4). Generally, we believe these graphical depictions of our data have many caveats and were included at the suggestion of Reviewer 3. However, we are cautious to make too many conclusions from this figure alone. The mean distance between and thickness of stress lines (right graph) does not fully convey the variation that is very much present in our sample (see left graph). This variation is important to consider and thus while the mean has some weight, we are weary of leaning heavily on means to make conclusions in this particular dataset.

We do describe stress lines in South African specimens, but as stated in lines 77-79 of the previous submission, "South African specimens typically display a single stress mark followed by regular growth that constitutes the majority of the growth record of this population". What is now Figure 3 displays the order and pattern of stress lines as they occur along the tusk and thus, we believe it is a more robust measure for conclusions than Figure 4. The actual patterns in South African tusks demonstrate a short episode of metabolic stress that is expected for any vertebrate and gives no indication of sustained reduction of metabolic activity as would be expected in episodes of torpor. In sum, we are not prepared to suggest that South African specimens show evidence of torpor and feel that doing so goes beyond what the data can reasonably conclude.

The ellipses shown in Figure 4 are data ellipses at 95% confidence intervals for the variation in each data set, not "an artificial range of variability" as Review 3 suggests. No changes were made.

We apologize for the typo of Reviewer 3's name in his citation! Thank you for catching this and changes have been made.

Reviewer 3 suggested adding a sentence highlighting his previous work on dentine growth marks recorded in the tusks of *Diictodon*. We have not done this for two reasons. First, Thackeray's work was already cited (Ref. 20). Second, it does not merit calling out more prominently as his stratigraphic results don't directly bear on the geographic comparison being tested here.

We agree that there should be a more comprehensive understanding of the unit of measurement these growth marks represent. However, that is far outside the scope of this project, as it would require growth experiments across a wide range on non-mammalian vertebrates. The methods section already addresses the fact that there is only a single non-mammalian test of time represented by these lines (Erickson 1996) and that Erickson's results are insufficient to assign a credible measurement of time to our data set.

Thank you for finding the error in our Rey (2017) citation!